# Effects of Framed Mobile Messages on Beliefs, Intentions, Adherence, and Asthma Control: A Randomized Trial

**DOI:** 10.3390/pharmacy12010010

**Published:** 2024-01-08

**Authors:** Ruth Jeminiwa, Kimberly B. Garza, Chiahung Chou, Ana Franco-Watkins, Brent I. Fox

**Affiliations:** 1Department of Pharmacy Practice, Thomas Jefferson University, 901 Walnut Street, Philadelphia, PA 19107, USA; rnj102@jefferson.edu; 2Health Outcomes Research and Policy, Harrison College of Pharmacy, Auburn University, Auburn, AL 36849, USA; kbl0005@auburn.edu (K.B.G.);; 3College of Arts and Sciences, University of Kentucky, 202 Patterson Office Tower, Lexington, KY 40506, USA

**Keywords:** mHealth, message framing, medication adherence, inhaled corticosteroids, asthma

## Abstract

We aimed to examine the effects of framed mobile messages (messages emphasizing losses or gains because of a behavior) on young adults’ beliefs about their daily Inhaled Corticosteroids (ICS), intentions to take their ICS, adherence, and asthma control. College students (18–29 years) who owned a mobile phone and had a diagnosis of asthma with a prescription for an ICS were recruited. Participants (*n* = 43) were randomized to receive either gain- or loss-framed mobile messages three times per week for eight weeks. Engagement rates with messages were calculated. Outcomes included beliefs, intentions, adherence, and asthma control. Data collection was performed at baseline, week 4, and week 8. Mixed-design ANOVA assessed whether outcomes improved differentially from baseline to week eight between gain- and loss-framed groups. Twenty-two participants were randomly assigned to the gain-framed group and 21 to the loss-framed group. There was a 100% retention rate. The engagement rate with the text messages was 85.9%. There was a significant difference in participants’ intentions to take medication and asthma control from baseline. There were no significant changes in other outcomes from baseline. There was no difference in changes in all outcomes between participants receiving gain- versus loss-framed messages. Framed mobile messages improved young adults’ asthma control and intentions to take their medication as prescribed.

## 1. Introduction

Asthma poses a significant clinical and economic burden to society, with approximately USD 80 billion per year spent on asthma-related medical expenses in the United States [1]. According to the Centers for Disease Control and Prevention, more than 1.6 million Americans visited the emergency department because of asthma in 2018 [2]. Patients with asthma typically have lifestyle limitations and are restricted in their ability to participate in certain physical activities, such as sports or going out with friends [3,4]. Inhaled corticosteroids (ICS), a mainstay treatment for persistent asthma, may help to reduce the morbidity and mortality associated with uncontrolled asthma. However, the rates of adherence to ICS are low. Adherence to ICS is less than 50% for adolescents and adults [5]. In one study, half of the patients with asthma refilled a 30-day ICS prescription only once in 12 months [6].

Young adults are at an increased risk of nonadherence to inhaled corticosteroids due to the unique challenges of emerging adulthood and being new to disease self-management [7,8]. Illness management including adherence to medications tends to decline during emerging adulthood, with young adults more likely to receive ambulatory care from emergency departments compared to adolescents [8,9]. Young adulthood is a developmental period that is marked by independence, decreased parental support and oversight, identity exploration, instability, self-focus, feeling in between adolescence and adulthood, the development of new social networks, increased risk-taking behaviors, and a low perception of risk [7,10,11]. It is also a period when individuals often assume responsibility for their own healthcare [8]. Young adults often face economic barriers such as lack of insurance or inability to afford medications [10,12,13]. This demographic has also reported difficulty in getting time off of school or work to make appointments with healthcare providers or refill their medications [10]. Additionally, young adults have a need to fit in with their peers rather than stand out [14]. As a result, they may forgo taking their asthma medication in order to be accepted [14]. Furthermore, negative perception of asthma medications and concerns about side effects and taste are also barriers to medication adherence reported by young adults [10,14,15].

Intentional nonadherence is critical in the management of asthma because patients have been found to intentionally interrupt their ICS based on waning symptoms or perceptions of need [16]. Furthermore, patients have reported their dislike of medications and fatigue of long-term medication taking as reasons for nonadherence to inhaled corticosteroids [15]. While many interventions have been implemented to improve adherence to ICS, very few address intentional nonadherence [17,18]. Message framing, the art of emphasizing losses or gains associated with engaging in a behavior, has been used to promote the adoption of various healthful behaviors including vaccinations, smoking cessation, use of sunscreens, mammography or breast self-examination, and other types of health screening [19,20,21,22].

According to Rothman and Salovey, the persuasiveness of gain- versus loss-framed messages depends on whether the behavior is focused on the discovery (detection behavior) or prevention (preventive behavior) of a health problem [23]. For preventive behaviors such as medication taking, the gain frame is expected to be more persuasive because people are risk-averse when considering gains. Conversely, detective behaviors such as health screenings risk identifying unfavorable findings, and therefore, loss-framed messages may be more persuasive [24]. This is based on the prospect theory, which suggests that when considering loss, individuals are more risk-taking, and when considering gains, individuals are risk-averse [25]. Therefore, emphasizing gains should be more advantageous in the context of the gains available through illness prevention behaviors such as medication adherence.

However, little is known about the effects of message framing on medication adherence. An intervention specifically targeted at young adults that utilizes framed mobile messages to address intentional nonadherence may effectively manage asthma control in this population. We hypothesize that presenting health information as a gain rather than a loss will lead to (1) more positive beliefs about inhaled corticosteroids, (2) greater intentions to take inhaled corticosteroids as prescribed, (3) greater medication adherence, and (4) better asthma control. We also hypothesize that all participants will have improved beliefs, intentions, adherence, and asthma control at the end of the study, regardless of the message frame. 

## 2. Materials and Methods

We conducted a parallel-arm randomized trial and collected data at baseline, week 4, and week 8. A computer program randomly assigned eligible participants to the gain-frame (group 1) or loss-frame (group 2) at a 1:1 ratio. Participants were blinded to treatment allocation. Participants were not informed of the group they belonged to throughout the study. The primary authors’ institutional review board approved the study.

Participants and recruitment: Eligible participants were college students enrolled at a university in the Southeastern United States, aged 18 to 29 years old, who could read and write English, had a diagnosis of asthma, and indicated possession of a prescription for an inhaled corticosteroid. The age bracket of 18 to 29 was used because this is the age bracket of young adults [26]. Possession of a prescription for an inhaled corticosteroid was confirmed by asking the students about the name of their medication and the dosage. Participants had to possess a smartphone and be willing to receive text messages during the study. All participants were recruited on a rolling basis between August and September 2019 via flyers distributed around campus and an email sent to all students. The study participants read and signed an informed consent form before the study started. They received Amazon gift cards to encourage their participation and prevent attrition. Specifically, USD 10 was provided at baseline after completing the baseline survey. A draw for a chance to receive a USD 10 Amazon gift card was performed at the end of week four, with the chance of being drawn equivalent to one out of ten. Finally, a USD 15 Amazon gift card was provided to all participants upon completing the study at the end of week eight. 

Intervention: The intervention consisted of gain- and loss-framed text messages [Appendix A]. The messages were developed under the guidance of a decision science expert with input from an individual with asthma after performing a systematic review of qualitative studies describing the experience of living with asthma as a young adult. The gain- and loss-framed messages were sent to participants randomized to groups 1 and 2, respectively, via text messages. The framing intervention was not explained to participants; therefore, they were unaware that they received either a loss- or gain-framed intervention. Text messages were sent thrice weekly between 10 am and 7 pm via a third-party texting platform, EZ texting [26]. Text messages were only delivered on weekdays as young adults do not desire text messages to be sent on weekends [27]. Participants were requested to acknowledge the receipt of each message by replying with an “R”. Texting began in August and ended in October 2019 on a rolling basis as participants were recruited.

### 2.1. Measures and Data Collection

Demographic Information: participants’ demographic information, including gender, age, ethnicity, and race, was collected during the baseline survey administered via Qualtrics. 

### 2.2. Study Outcomes and Measures

Beliefs: Beliefs were measured with the Beliefs about Medications Questionnaire (BMQ-Specific) [28]. BMQ-Specific consists of two scales comprising five items each. The first scale assesses beliefs about the necessity of prescribed medications, while the second scale assesses concerns about prescribed medications based on beliefs about the danger of dependence, long-term side effects, and disruptive effects of medications. Beliefs were calculated as the difference between necessity and concerns scores, with a possible range of −20 to 20 [29]. The higher the score, the greater the perceived necessity of the medication compared to concerns. For example, a person scoring 6 points has a greater perceived necessity of the medication compared to concerns than someone scoring 3 points. Beliefs were measured at baseline and the end of week eight.Intentions: The sum of participants’ responses to three items (“I intend to take my inhaled asthma medications as prescribed”; “I will take my daily inhaled asthma medication as prescribed”; and “I will always take my inhaled asthma medications as prescribed”) was used to measure intentions [30]. These items were scored on a five-point Likert-type scale from strongly disagree (1) to strongly agree (5). Therefore, the minimum score was 3, indicating low intentions, and the maximum score possible, indicating high intentions, was 15. Intentions were measured at baseline and the end of week eight.Medication Adherence: The Medication Adherence Report Scale for Asthma (MARS-A) [31] was used to elicit participants’ adherence to their controller medications (ICS or ICS combinations) at baseline, week four, and week eight. Each item is rated on a five-point Likert-type scale, with higher scores indicating greater adherence. Adherence was measured on a continuous scale as recommended by the literature, rather than as a dichotomous division into adherent/nonadherent categories [32]. A previous study also used the MARS-A instrument in a similar manner [29]. The mean scores were used in this study. The mean score ranges from 1 to 5 points, with higher scores indicating higher adherence to medications. This scale has demonstrated good test–retest reliability (*r* = 0.65, *p* < 0.001) and correlates well with electronic adherence (*r* = 0.42, *p* < 0.001) [31].Asthma: The ACT questionnaire, a valid and reliable self-report [33], was used in measuring participants’ asthma control at baseline, week four, and week eight. The ACT consists of five items, which are rated by the individual whose asthma control is to be measured. It measures asthma control by asking questions pertaining to asthma symptoms that occurred in the past four weeks. The response set ranges from All of the time (1) to None of the time (5). The total scores for the ACT range from 5 to 25, with individuals scoring 19 and less considered uncontrolled [34]. An example ACT item is, “In the past four weeks, how much of the time did your asthma keep you from getting as much done at work, school, or at home?” The ACT has demonstrated high internal consistency (Cronbach’s’ α = 0.85) and significant correlations with specialists’ rating of asthma control (*r* = 0.52, *p* < 0.001) [33]. A minimally important difference (MID) of 3 reflects the smallest difference in score with a clinically significant change.

### 2.3. Other Variables Measured

Social desirability bias: Social desirability is the need of individuals to obtain approval by responding in a culturally appropriate manner [35]. Since self-reporting is susceptible to social desirability bias, the short form of the Marlowe Crown Scale was administered to identify responses that were made in a socially desirable manner [36].Type of nonadherence: Two items from MARS–A were used in measuring the type of nonadherence as reported by a previous study [37,38]. Specifically, individuals who indicated agree or strongly agree to either of the following sentences were categorized as intentionally nonadherent: “I alter the dose” and “I decide to miss out a dose”.

Sample size: To detect an asthma control effect size of 0.25 at a significance level of 5% and a power of 80%, a total of 86 participants were required. A conservative effect size of 25% was chosen because existing studies do not provide effect size estimates. Sample size calculations were performed using the G*power statistical power analysis program [39]. 

Statistical analysis: Descriptive statistics were performed for all variables. The correlation between the social desirability bias score and all study variables was assessed. Exploratory factor analysis was performed to determine the construct validity of the intention scale, the only scale not previously validated. Cronbach’s alpha coefficient was also calculated to determine the internal consistency of the intention scale. The average beliefs and intention scores measured at baseline and week eight were compared across groups using the two-way mixed ANOVA. We controlled for the type of nonadherence because we expected the type of nonadherence (intentional versus unintentional) to impact our findings. The average MARS-A and ACT scores were compared across groups at baseline, week four, and week eight using two-way mixed ANOVA. Changes in all parameters from baseline, irrespective of group, were also assessed. Using the chi-square test, we also compared the proportion of individuals achieving a MID of 3 at week four and week eight. The foundation which funded this study played no role in the study design, execution, or reporting. All statistical analyses were performed using SPSS version 23 [40].

## 3. Results

### 3.1. Participants

Eighty-one participants were screened after they reached out to the primary investigator indicating an interest to participate in the study. Of the 81 participants screened, 43 met the eligibility criteria and were willing to participate in the study. All 43 students were recruited for the study (Figure 1) and contributed data at baseline, week 4, and week 8. Table 1 displays the characteristics of the participants. Most participants were female (60%), non-Hispanic (93%), white (90.7%), were diagnosed with asthma as a child (83.7%), had not missed school or work in the past month due to asthma (81.4%), and had never smoked (88.4%). The mean age of participants was approximately 21. The mean asthma control was approximately 20 (5 = poor asthma control, 25 = completely controlled asthma). The mean adherence was 3.52, with the maximum possible score being 5.0. At baseline, most participants believed in the necessity of their ICS for maintaining health. The present sample had a moderate amount of social desirability bias (mean score = 6.14). In a bivariate analysis, two variables (concerns measured at week eight and asthma control measured at week four) significantly correlated with social desirability bias.

### 3.2. Evaluation Outcomes

Beliefs: There was no significant difference in beliefs between participants receiving gain- versus loss-framed messages. Changes in beliefs for all participants from baseline to the end of the study were also not significant, even after controlling for the type of nonadherence. The mean scores across different time points are shown in Table 2. The two-way ANOVA results are shown in Table 3.

Intentions: Exploratory factor analysis showed all three items loaded on one factor, intentions, with an eigenvalue greater than 1. This factor explained 81.87% of the total variance. The intention scale had a high internal consistency with a Cronbach’s alpha of 0.886. The two-way mixed ANOVA result (Table 4) indicates no statistically significant interaction between the intervention and time. There was no statistically significant difference in intentions from baseline among participants receiving gain- versus loss-framed messages. The mean intention scores of participants at different time points are shown in Table 2. However, the main effect of time was significant after controlling for the type of nonadherence, indicating that there was a statistically significant difference in the level of intentions of all participants from baseline to the end of study. The covariate type of nonadherence was statistically significantly related to time (F (1, 38) = 7.788, *p* = 0.008). 

Medication adherence: The mean adherence score of participants across the different time points is shown in Table 2. There was no significant difference in adherence levels based on whether participants received a gain- or loss-framed message. Furthermore, there was no significant difference in adherence levels from baseline across week 4 and week 8 (Table 5).

Asthma control: There was no significant interaction between the experimental condition (gain- vs. loss-framed messages) and time of measurement (baseline, week 4, and week 8) of asthma control. There was no significant difference in asthma control based on whether a participant received gain- versus loss-framed messages. Participants’ asthma control differed in week four and at the end of the intervention compared to baseline. Participants had greater asthma control eight weeks after the intervention compared to four weeks after the intervention. Also, they had greater asthma control eight weeks after the intervention compared to baseline. The mean asthma control scores across all time points are shown in Table 2. The two-way ANOVA results are displayed in Table 6.

The proportion of individuals with a minimally important difference of three is displayed in Figure 2. Among the present sample, participants had greater asthma control after the framing intervention, whether they received gain- or loss-framed messages. 

### 3.3. Study Retention and Engagement with Text Messages 

There was 100% retention of participants who enrolled in the study. Also, all participants contributed data at the baseline, week four, and the end of the study at week eight. A total of 1032 text messages were successfully delivered to the participants. The participants’ engagement with the text messaging intervention was 85.9%, as assessed by their response to each text message received during the intervention. According to the texting platform, there were no errors or failures in message delivery, indicating a 100% transmission rate.

## 4. Discussion

This study assessed the effects of framed mobile messages on college students’ beliefs, intentions to take their ICS, adherence, and asthma control. There were improvements in participants’ intentions to take their ICS and asthma control at the end of the study compared to baseline, regardless of the message framing. There were no significant differences in participants’ beliefs and adherence at the end of the study compared to baseline. Our study found no relative advantage of gain- versus loss-framed messages on all assessed variables—beliefs about ICS, intentions to take ICS, medication adherence, and asthma control.

Our study found no relative advantage of gain- versus loss-framed messages on participants’ beliefs about their ICS. Additionally, there was no significant difference in participants’ beliefs at the end of the study compared to baseline. There is a risk of uncovering an unfavorable condition in disease detection behaviors [41]. Therefore, emphasizing the loss of not performing an advocated behavior is more persuasive because people are more risk-seeking when considering risks [42]. The gain frame is expected to be more persuasive for preventive behaviors because people are risk-averse when considering gains; therefore, focusing on gains instead of loss is more persuasive [41]. For example, in a study conducted by Rivers and colleagues, the effectiveness of framed messages in persuading women to obtain a pap smear test was contingent on how the behavior was framed [43]. Gain-framed messages were more effective if the pap smear test was framed as a prevention behavior. Loss-framed messages were more effective if the pap smear test was framed as a detection behavior. 

Despite the significant results obtained by these previous studies on the effectiveness of gain-framed messages in promoting positive beliefs, a recent meta-analysis reported findings consistent with our study. They found no significant framing effects on attitudinal beliefs among studies examining a prevention behavior [44]. In our study, participants were asked to acknowledge the receipt of each message as a proxy to track participant engagement. At the end of the study, a participant mentioned that he did not agree with some of the messages and would have loved an opportunity to express his opinion. Therefore, assessing participants’ level of agreement with framed messages may have provided insight into whether this variable modifies framing effects on beliefs. Furthermore, Rothman and Salovey suggest that a participant’s acceptance of a message is crucial in framing’s effectiveness [24]. In the present study, framing had no effect on participants’ beliefs about their ICS. Of note, participants’ agreement with the messages was not assessed. Future studies should assess the relevance of agreement with message effectiveness. The use of eHealth interventions may provide an opportunity for user feedback. Interestingly, eHealth interventions have yielded positive outcomes with asthma control and disease management behaviors [45,46].

In the current study, participants had high intentions to take their medications as prescribed at baseline and at the end of the study. Before controlling for the type of nonadherence, the initial analysis did not find a significant difference in intentions between participants receiving gain- or loss-framed messages. We also did not find a significant difference in intentions from baseline across both groups at the end of the study. After controlling for the type of nonadherence, framing had significant effects on participants’ intentions to take medications as prescribed. Irrespective of whether participants received gain- or loss-framed messages, their intentions increased from baseline, and the increment was statistically significant. It is possible that the presence of individuals who had intentional nonadherence and those who had unintentional nonadherence in the same sample served as a confounder in our initial analyses. There is mixed evidence on the effects of framed messages on intentions to perform illness prevention behaviors. Abhyankar et al. suggested that loss-framed messages were more effective in increasing intentions to obtain the Mumps, Measles, and Rubella vaccine [47]. Meanwhile, Detweiler et al. found that gain-framed messages were more effective in promoting intentions in another disease prevention behavior (the utilization of sunscreens) [19]. A recent meta-analysis of the effects of framed messages on illness preventive behaviors did not find a significant effect of framing on intentions [44]. This is consistent with our findings before controlling for the type of nonadherence. Therefore, we speculate that the presence of unknown confounders, such as the type of nonadherence in the case of medication adherence, could explain why framing effects on intentions may not have been detected in other illness prevention studies.

In our study, we considered adherence to inhaled corticosteroids a preventive behavior because this medication-taking behavior aims to prevent asthma exacerbations and ensure asthma control. Based on the recommendation of Rothman et al. and findings from a review conducted by Gallagher et al. [24,44], we hypothesized that gain-framed messages would be more persuasive in encouraging young adults to take their ICS as prescribed. In the present study, there was no advantage to gain framing. Although there was a slight improvement in adherence over time, it was not significant, regardless of the message framing. Of note, the acceptability of the framed messages to target population members was not assessed. Future studies should assess the effect of message acceptability on framing effects. Furthermore, our participants were adherent to their ICS at baseline, making it difficult to detect the impact of the intervention. Future studies should target a nonadherent patient population.

Studies have been conducted investigating the differential effects of gain- versus loss-framed messages to promote a range of illness prevention behaviors, including exercise adherence and smoking cessation. In a study promoting exercise adherence among patients entering a cardiac rehab program, gain-framed messages resulted in greater exercise participation than control [48]. Furthermore, a recent meta-analysis of the effects of health message framing on behavior found that gain-framed messages promoting illness prevention behaviors were more persuasive than loss-framed messages [44]. No published studies have assessed the relative advantage of gain- versus loss-framed messages on medication adherence.

Among the sample in our study, the average asthma control at baseline was 20, indicating that participants were controlled but with room for improvement, since the minimum score for control is 20. We did not find an advantage from gain framing over loss framing in improving asthma control among participants in the study. The reason behind this finding is not clear. However, we found that irrespective of the type of framed message received by participants, there was a significant improvement in asthma control at the end of the study compared to the baseline. According to our data, the framed messages explained about 7.5% of the variance in asthma control. This finding may have also been due to the effect of text messaging regardless of framing, since text messages alone have been found to impact behavior change and improve health outcomes [49]. To our knowledge, no published study has investigated the effects of framing on asthma control. We expected an increase in medication adherence to improve asthma control. Albeit not significant, we did observe an increase in medication adherence. Furthermore, it is possible that the little improvement in medication adherence was sufficient to lead to improved asthma control among the present sample. Newmann and colleagues assessed the impact of community health workers on adherence using two types of adherence measures (30-day and 5-day recall) [50]. They observed significant improvements in adherence based on the 30-day recall measure. However, they found no significant difference in adherence as measured with the 5-day recall measure. Our method of assessing adherence may have played a role in the difficulty in detecting small but significant changes in adherence. It would be interesting to also assess asthma control through the frequency of use of short-acting beta-2-agonists by participants. Future studies should consider collecting these data too. Other explanations for our significant improvements in asthma control without a significant improvement in adherence could be that our intervention impacted other self-care behaviors, such as decreased smoking or decreased exposure to triggers, which were not assessed in the study.

### Limitations

This study has several limitations that should be considered. The sample size was small, which did not provide adequate power to assess the differential effects of gain versus loss framing on outcome variables. In addition, our sample was focused on young adults with asthma enrolled in college and may not generalize to other young adults that may not be attending college. In this study, we also relied on self-reporting to measure participants’ beliefs, intentions, adherence, and asthma control. Self-reporting is subject to social desirability bias, and we found moderate social desirability in our study. However, it only correlated significantly with two measures: concerns about medication measured at the end of the study and asthma control measured at the study’s midpoint (week four). Self-reporting is also vulnerable to recall bias. It is possible that using more objective measures may have yielded a different result. Another limitation in the framing interventions is that a diagnosis of asthma and possession of an ICS was via self-reports. There was no way to confirm participants’ self-reports other than asking for the name of the ICS that the participants were taking and the prescribed dosage.

Furthermore, data from the texting platform suggest over 85% participant engagement. However, there is no way to ascertain that participants were the ones acknowledging the receipt of messages. Finally, this study did not account for the seasonal variation in asthma. There is evidence that asthma symptoms peak for young adults in the US in September through November, which happens to be the period within which this study was conducted. Achieving better asthma control during the peak period for asthma symptoms further suggests a positive impact of the intervention.

## 5. Conclusions

In this study, framed mobile messages delivered via SMS improved young adults’ intentions to take their medication as prescribed and their asthma control. Further studies with a control group are needed to support causality between framing, intentions to take medications, and asthma control. A 100% retention rate and over 85% engagement rate suggest that young adults are comfortable receiving text messages promoting adherence to ICS. The findings from this study have important implications for health professionals and other stakeholders interested in helping young adults with asthma achieve better control of their asthma.

## Figures and Tables

**Figure 1 pharmacy-12-00010-f001:**
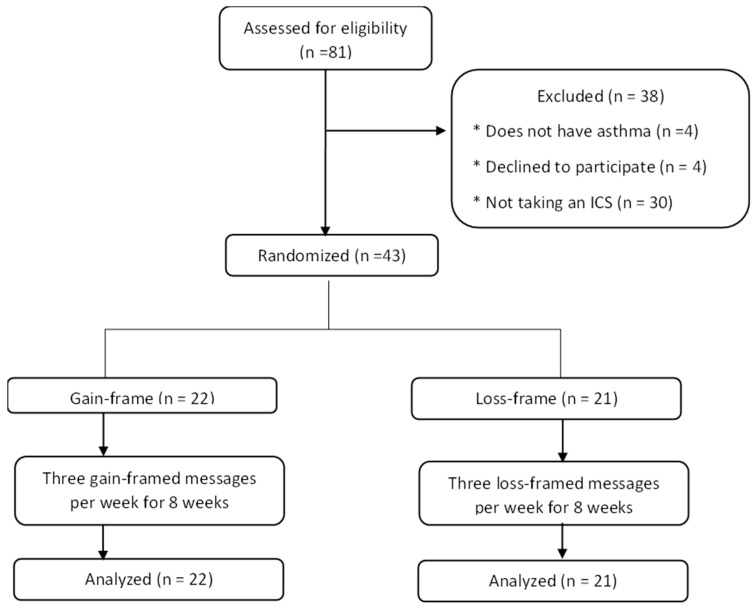
CONSORT flow diagram.

**Figure 2 pharmacy-12-00010-f002:**
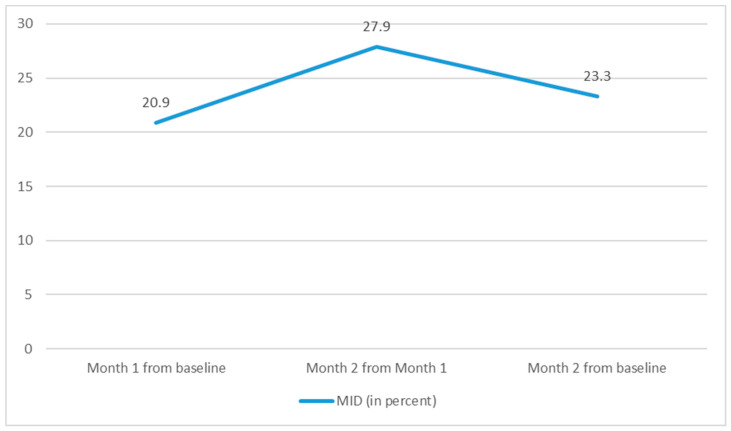
The proportion of participants achieving a minimally important difference or MID (the smallest difference in score with a clinically significant change) of three across time points.

**Table 1 pharmacy-12-00010-t001:** Participants’ characteristics at baseline.

Categorical Variables ^a^	*n* (%)			*p*-Value
	Total (*n* = 43)	Gain-Frame (*n* = 22)	Loss-Frame (*n* = 21)	
**Gender** ** Male** ** Female**	14 (32.6) 29 (67.4)	8 (36.4) 14 (63.6)	6 (28.6) 15 (71.4)	0.586
**Ethnicity** ** Hispanic** ** Non-Hispanic**	3 (7.0) 40 (93.0)	2 (9.1) 20 (90.9)	1 (4.8) 20 (95.2)	0.578
** Race** ** Black or African American** ** White** ** Others**	3 (7.0) 39 (90.7) 1 (2.3)	2 (9.1) 20 (90.9) 0	1 (4.8) 19 (90.5) 1 (4.8)	0.513
**Smoking status** ** Formerly smoked** ** Currently smokes** ** Never smoked**	5 (11.6) 0 38 (88.4)	5 (22.7) 0 17 (77.3)	0 0 21 (100)	*0.048*
**Length of asthma diagnosis** ** 1 to 5 years ago** ** Over 5 years ago** ** Diagnosed as a child**	5 (11.6) 2 (4.7) 36 (83.7)	4 (18.2) 1 (4.5) 17 (77.3)	1 (4.8) 1 (4.8) 19 (90.5)	0.365
**Average duration of missing work or school in the past month** ** Did not miss work or school** ** 1–3 days** ** 4–6 days** ** More than 7 days** ** Not in school or working in past month**	35 (81.4) 4 (9.3) 0 0 4 (9.3)	18 (81.8) 2 (9.1) 0 0 2 (9.1)	17 (81.0) 2 (9.5) 0 0 2 (9.5)	0.997
**Continuous variables ^b^**	Mean (SD)			
**Age, years**	20.86 (2.07)	20.36 (1.50)	21.38 (2.46)	0.107
**Asthma control**	19.77 (3.18)	20.18 (2.92)	19.33 (3.45)	0.389
**Medication adherence**	3.52 (0.83)	3.47 (0.85)	3.57 (0.82)	0.701
**Concerns about ICS**	10.91 (4.24)	10.45 (4.34)	11.38 (4.19)	0.481
**Perceived necessity of ICS**	16.93 (4.9)	15.91 (5.44)	18.00 (4.12)	0.165
**Necessity-concerns differential**	6.02 (5.92)	5.45 (6.67)	6.62 (5.10)	0.525
**Intentions to take ICS ^c^**	12.81 (2.63)	12.18 (3.17)	13.47 (1.75)	0.166
**Social desirability bias**	6.14 (1.57)	5.91 (1.60)	6.38 (1.53)	0.330

^a^ for categorical variables, the chi-square test was used in comparing the two groups. ^b^ for continuous variables, an independent *t*-test was used in the comparisons. ^c^ computed with Mann–Whitney due to outliers. Significantly different values are marked in italics.

**Table 2 pharmacy-12-00010-t002:** Outcome score of participants receiving gain- vs. loss-framed messages.

Variable		Mean Scores (SD)		
Beliefs	**Timepoint**	**All**	**Gain-Framed**	**Loss-Framed**
	Baseline	6.02 (5.91)	5.45 (6.67)	6.62 (5.10)
	Week 8	5.00 (6.78)	4.09 (7.46)	5.90 (6.02)
Intentions	Baseline	13.12 (2.14) *+	12.18 (3.17)	13.48 (1.75)
	Week 8	13.37 (1.73) *+	12.73 (2.62)	13.14 (2.65)
Adherence	Baseline	3.52 (0.83)	3.47 (0.85)	3.57 (0.82)
	Week 4	3.55 (0.80)	3.40 (0.80)	3.70 (0.79)
	Week 8	3.57 (0.87)	3.52 (0.83)	3.62 (0.93)
Asthma control	Baseline	19.77 (3.18) *	20.18 (2.92)	19.33 (3.45)
	Week 4	19.95 (3.45) *	20.50 (3.35)	19.38 (3.54)
	Week 8	20.93 (2.63) *	21.23 (2.20)	20.62 (3.04)

* significantly different values are marked with asterisks. + adjusted for type of nonadherence (level of intentional nonadherence was used as a covariate in a 2-way mixed ANOVA).

**Table 3 pharmacy-12-00010-t003:** Two-way ANOVA result for beliefs.

*Source*	*Sum of Squares*	*Degree of Freedom*	*Mean Square*	*F-Test*	*p-Value*	*Partial Eta squared*	*Observed Power*
*Time (baseline, week 4, week 8)*	*23.195*	*1, 41*	*23.195*	*2.385*	*0.130*	*0.055*	*0.326*
*Group (loss* vs. *gain)*	*47.654*	*1, 41*	*47.654*	*0.662*	*0.421*	*0.016*	*0.125*
*Interaction*	*2.265*	*1, 41*	*2.265*	*0.233*	*0.632*	*0.006*	*0.076*

**Table 4 pharmacy-12-00010-t004:** Two-way ANOVA result for intentions *.

Source	Sum of Squares	Degree of Freedom	Mean Square	F-Test	*p*-Value	Partial Eta Squared	Observed Power
Time (baseline, week 4, week 8)	11.267	1, 38	11.267	8.585	0.006	0.184	0.815
Group (loss vs. gain)	7.349	1, 38	7.349	1.791	0.189	0.045	0.257
Interaction	1.108	1, 38	1.108	0.844	0.364	0.022	0.146

* controlled for type of nonadherence.

**Table 5 pharmacy-12-00010-t005:** Two-way ANOVA result for medication adherence.

Source	Sum of Squares	Degree of Freedom	Mean Square	F-Test	*p*-Value	Partial Eta Squared
Time (baseline, week 4, week 8)	0.057	1.58, 64.94	0.036	0.204	0.764	0.005
Group (loss vs. gain)	0.878	1, 41	0.878	0.481	0.492	0.012
Interaction	0.274	1.58, 64.94	0.173	0.983	0.363	0.023

**Table 6 pharmacy-12-00010-t006:** Two-way ANOVA results for asthma control.

Source	Sum of Squares	Degree of Freedom	Mean Square	F-Test	*p*-Value	Partial Eta Squared
Time (baseline, week 4, week 8)	33.76	2, 82	16.88	3.34	0.040	0.075
Group (loss vs. gain)	23.76	1, 41	23.76	1.25	0.269	0.030
Interaction	1.403	2, 82	0.702	0.14	0.871	0.003

## Data Availability

The data presented in this study are available on request from the corresponding author.

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
