# Peer review of "Effects of Framed Mobile Messages on Beliefs, Intentions, Adherence, and Asthma Control: A Randomized Trial"

_pharmacy, 2024, doi:10.3390/pharmacy12010010_

Round 1

Reviewer 1 Report

Comments and Suggestions for Authors

Will the authors consider the fact that the sample selection was not the best? The people who signed up for the study are adherent and with good asthma control and mainly female. Are females more adherent than males especially in younger generation? Would the study had different results if the sample were selected from non-adherent youngsters from a clinic?

Author Response

Reviewer's comment. Will the authors consider the fact that the sample selection was not the best? The people who signed up for the study are adherent and with good asthma control and mainly female. Are females more adherent than males especially in younger generation? Would the study had different results if the sample were selected from non-adherent youngsters from a clinic?

Our response: 

Thank you for your question about the sample selection. The focus of our research was young adults enrolled at our university. An opportunity was provided for every student with asthma meeting the eligibility criteria to enroll. It is possible that those who enrolled did so because they recognize the importance of adherence and were mostly adherent themselves. We acknowledged the concern about the high level of adherence in the discussion section using the sentence below.

"Furthermore, our participants were adherent to their ICS at baseline making it difficult to detect the impact of the intervention. Future studies should target a nonadherent patient population."[Line 393 - 395]

Reviewer 2 Report

Comments and Suggestions for Authors

Thank you very much for giving an opportunity to review the present manuscript. The authors examined the effects of gain-framed mobile messages on young adults' beliefs about their daily ICS, intentions to take their ICS, adherence, and asthma control. They found that there was no difference in changes in all outcomes between participants receiving gain- versus loss-framed messages, and framed mobile messages improved young adults' asthma control and intentions to take their medication as prescribed.

Although this is a very interesting study on behavioral psychology, and important findings have been obtained, there are some improvements that should be made before publication.

1. Since the number of samples is small, shouldn't it be necessary to analyze the median value instead of the average value?

2. In tables, significantly different values are marked in italics, but it is difficult to understand, so please use another notation, such as an asterisk.

3. "Figure 2" is not found in the text.

4. It may be possible to see a clearer effect of gained-frame mobile messages on medication adherence by targeting patients with poor adherence. How about mentioning this in the discussion?

Author Response

Reviewer's comment:

  1. Since the number of samples is small, shouldn't it be necessary to analyze the median value instead of the average value?

Our response:  Thank you for your feedback. We did not use the median values in our analysis because there were no observed skew, kurtosis, or outliers in the data.

Reviewer's comment:

  1. In tables, significantly different values are marked in italics, but it is difficult to understand, so please use another notation, such as an asterisk.

Our response: Thank you, we revised as suggested

Reviewer's comment: "Figure 2" is not found in the text.

Our response: Figure 2 is on page 9, line 271 – 273.

Reviewer's comment: 4. It may be possible to see a clearer effect of gained-frame mobile messages on medication adherence by targeting patients with poor adherence. How about mentioning this in the discussion?

Our response: Great feedback. Thank you. We have revised as suggested (lines 393-395).

Reviewer 3 Report

Comments and Suggestions for Authors

Thank you for allowing me to review this work.

This is a study, from my point of view, well designed and with a solid hypothesis.

After reading it carefully, I consider that it presents a significant problem in terms of its execution, since it was not possible to reach the minimum size calculated a priori. The participation rate was 50%, a rate that was too low to detect differences and to be able to demonstrate the study hypothesis, and consequently, the main objective.

Perhaps a good strategy would be to calculate the sample size based on non-inferiority of one strategy over the other.

On the other hand, the results are interesting and well discussed, in turn issuing good conclusions based on the results.

The structure and content of the article is good.

I consider that currently, this article does not meet the requirements for publication in this journal, but I could focus it on other journals in our group, mdpi, such as the Journal of Respiration.

Thank you so much

Author Response

Reviewer's comment:  Thank you for allowing me to review this work.
This is a study, from my point of view, well designed and with a solid hypothesis. After reading it carefully, I consider that it presents a significant problem in terms of its execution, since it was not possible to reach the minimum size calculated a priori. The participation rate was 50%, a rate that was too low to detect differences and to be able to demonstrate the study hypothesis, and consequently, the main objective.
Perhaps a good strategy would be to calculate the sample size based on non-inferiority of one strategy over the other.

Our response: Thank you for the great feedback. There are no trials demonstrating the effectiveness of either message frame on improving patients’ asthma control. Therefore, we couldn’t utilize the non-inferiority strategy. D’Agostino RB, Non-inferiority trials: design concepts and issues – the encounters of academic consultants in statistics. Statistics in Medicine. 2003; 22:169-186.

Reviewer 4 Report

Comments and Suggestions for Authors

Dear Authors,

the study you presented is interesting and relevant. It refers to very current problem of asthma and its medication (incl. adherence).

I found a few issues that should be addressed in your study:

1. Message framing is not an internationally used and understood scientific term. For this reason, please briefly state its meaning in the Abstract and then describe it in details in the Introduction section.

The results would benefit from being presented in figures and tables rather than being described in the text. Their more detailed description could still be present in the Discussion section.

The references are provided in two different manners throughout the text. First part of the text has references in brackets, e.g. [1] etc., then they are in superscript numbers. Moreover, the list of references contains only 10 positions, whereas much more are cited in the text.

Author Response

Reviewer's comment: Dear Authors, the study you presented is interesting and relevant. It refers to very current problem of asthma and its medication (incl. adherence).

Our response: thank you

Reviewer's comment: I found a few issues that should be addressed in your study:
1. Message framing is not an internationally used and understood scientific term. For this reason, please briefly state its meaning in the Abstract and then describe it in details in the Introduction section.

Our response: We have included a definition in the abstract. We have also revised the definition in the introduction section of the paper (lines 68-69).

Reviewer's comment: The results would benefit from being presented in figures and tables rather than being described in the text. Their more detailed description could still be present in the Discussion section.

Our response: We have deleted textual information of the results and presented them in Tables as suggested. Thank you for the feedback.

Reviewer's comment: The references are provided in two different manners throughout the text. First part of the text has references in brackets, e.g. [1] etc., then they are in superscript numbers. Moreover, the list of references contains only 10 positions, whereas much more are cited in the text.

Our response: Thank you, the references have been revised and updated.

Reviewer 5 Report

Comments and Suggestions for Authors

To the authors:

Thank you for the opportunity to read and review the article titled “Effects of Framed Mobile Messages on Beliefs, Intentions, Adherence, and Asthma Control: A Randomized Trial” for the journal Pharmacy. This study has tested mobile text messages as an intervention to affect young adults’ use of ICE: adherence, intentions, beliefs, and asthma control. The correct use of asthma medications is of great importance for asthma management, and interventions attacking for example the correct use of inhalers and adherence is warranted. I really enjoyed reading the manuscript and believe it will be of interest to the readers of Pharmacy.

I suggest revisions to the manuscript before publication. Importantly, you need to include a complete reference list. I was not able to review your references.

I hope my suggestions and comments are helpful!

Introduction:

Very well written. However, there are references in the introduction that are missing in the reference list.

Method:

The title and abstract indicate that this is a randomized trial. However, I cannot find information about the randomization process and random sequence generation. Please, include information about the method used to generate the allocation sequence, and any measures to conceal allocation before or during enrolment.

What were the measures to minimize performance bias? Did the participants receive information about the hypothesis on gain vs loss framed messages? I guess the nature of intervention does not allow for full blinding, but you might want to include some information about your strategy. Describe all measures used, if any, to blind participants from knowledge of which intervention they received.

What were the measures to minimize detection bias? Describe all measures used, if any, to blind outcome assessment (researchers) from knowledge of which intervention a participant received. Provide any information relating to whether the intended blinding was effective.

About outcome measures:

·       BMQ: maybe add some information about how to understand the concern-necessity scale and differential (for example the results of N/C diff at 6.02 and 5.00, what do they mean, how can we interpret the result?).

·       Intentions: considering adding information on min-max range of possible results, and information on how to understand the results.

·       Adherence: considering adding information on min-max range of possible results, and information on how to understand the results. Did you define cut-off for “good” or “bad” adherence, or nonadherence?

·       It would be interesting to have information about participants use of other asthma medications. Did you collect data on SABA? If asthma control increased, they should use less SABA?

Results:

Comment on study sample: You calculated 86 participants were needed, however assessed only 81 for eligibility. Why did you stop at 81?

Comment on participants characteristics: what other categorical variables could influence the study results? Did you collect data on for example status (single, partner etc), living arrangement (with parents, alone, etc), socioeconomic indicators, the use of other (asthma) medications?

Consider presenting results of ANOVA in a table in addition to the text presentation.

Figure 2: consider explaining MID in the capture, making it convenient for the reader to understand the figure without referring to the manuscript text.

Discussion

Line 304-311: very interesting discussion, and might indicate a need for eHealth intervention that allows for engaging the participants, e.g., interactive responses? Are there other asthma mediation eHealth interventions targeting the young adults? Maybe you could extend your discussion including results from such studies? For example van Buul, A.R. et al.  2020. eHealth only interventions and blended interventions to support self-management in adolescents with asthma: A systematic review. Clinical eHealth;  49-62, ISSN 2588-9141, https://doi.org/10.1016/j.ceh.2020.06.001.

Line 334-343 and 371-375: Regarding adherence to asthma medications, I was wondering if the study participants had a challenge with adherence from baseline? Were they nonadherent? I guess it would be even more difficult detecting significant changes in adherence if the participants are adherent from baseline. Maybe future research should target a population with adherence challenges?

Comment on asthma control (from line 353): it would be interesting to have data on the use of SABA as an indicator of asthma control. Maybe add some discussion on additional clinical outcome measures for future research?

Limitations

With reference to previous comments on randomization and blinding: please add information on possible selection bias, performance bias, and detection bias. I recommend using RoB 2 as a guide for evaluation (https://methods.cochrane.org/bias/resources/rob-2-revised-cochrane-risk-bias-tool-randomized-trials).

References:

The reference list needs thorough revision; many references are missing.

Author Response

Reviewer 5

Reviewer's comment

Authors' response

Thank you for the opportunity to read and review the article titled "Effects of Framed Mobile Messages on Beliefs, Intentions, Adherence, and Asthma Control: A Randomized Trial" for the journal Pharmacy. This study has tested mobile text messages as an intervention to affect young adults' use of ICE: adherence, intentions, beliefs, and asthma control. The correct use of asthma medications is of great importance for asthma management, and interventions attacking for example the correct use of inhalers and adherence is warranted. I really enjoyed reading the manuscript and believe it will be of interest to the readers of Pharmacy.

I suggest revisions to the manuscript before publication. Importantly, you need to include a complete reference list. I was not able to review your references.

I hope my suggestions and comments are helpful!

Great feedback, thank you.

Introduction:

Very well written. However, there are references in the introduction that are missing in the reference list.

Thank you for your feedback. The references have been updated.

Method:

The title and abstract indicate that this is a randomized trial. However, I cannot find information about the randomization process and random sequence generation. Please, include information about the method used to generate the allocation sequence, and any measures to conceal allocation before or during enrolment.

Thank you. A computer program was used to randomly assigned participants to the two groups (Page 2, line 95). In the subsequent sentences, we added more information as suggested. Specifically, we stated that "A computer program randomly assigned eligible participants to the gain-frame (group 1) or loss-frame (group 2) at a 1:1 ratio. Participants were blinded to treatment allocation. Participants were not informed of the group they belonged to throughout the study."

What were the measures to minimize performance bias? Did the participants receive information about the hypothesis on gain vs loss framed messages? I guess the nature of intervention does not allow for full blinding, but you might want to include some information about your strategy. Describe all measures used, if any, to blind participants from knowledge of which intervention they received.

Thank you for the feedback on performance bias (differences in the care provided to comparison groups other than the intervention). The Cochrane Collaboration suggests blinding healthcare providers to prevent this risk. However, the recommendation does not apply to this intervention because it was delivered remotely. All participants received an intervention (either positively or negatively framed intervention). However, the framing intervention was not discussed with participants. We have added the sentence, "The framing intervention was not explained to participants; therefore, they were unaware that they received either a loss or gain-framed intervention." [line 122-123]

What were the measures to minimize detection bias? Describe all measures used, if any, to blind outcome assessment (researchers) from knowledge of which intervention a participant received. Provide any information relating to whether the intended blinding was effective.

About outcome measures:

·       BMQ: maybe add some information about how to understand the concern-necessity scale and differential (for example the results of N/C diff at 6.02 and 5.00, what do they mean, how can we interpret the result?).

The Cochrane Collaboration recommends blinding of outcome assessors to prevent detection bias. However, detection bias was not applicable to this intervention because participants self-reported their outcomes using a link to an online form sent as a text to their phone.

Great point; we have added the statement (lines 140-142), “The higher the score, the greater the perceived necessity of the medication compared to concerns. For example, a person scoring 6 points has a greater perceived necessity of the medication compared to concerns than someone scoring 3 points.” as suggested.

In Intentions: considering adding information on min-max range of possible results, and information on how to understand the results.

Great point, thanks. I have added additional information as suggested. I added the following sentence (lines 149-151) - “Therefore, the minimum score was 3, indicating low intentions, and the maximum score possible, indicating high intentions, was 15.

      Adherence: considering adding information on min-max range of possible results, and information on how to understand the results. Did you define cut-off for "good" or "bad" adherence, or nonadherence?

Thank you for your feedback. Adherence was measured on a continuous scale as recommended by literature, rather than as a dichotomous division into adherent/nonadherent categories. We added additional information to help readers interpret the result as suggested (lines 158-160). The additional information is “The mean score was used in this study. The mean score ranges from 1 to 5 points, with higher scores indicating higher adherence to medications.

 It would be interesting to have information about participants use of other asthma medications. Did you collect data on SABA? If asthma control increased, they should use less SABA?

This is a great point. However, we did not collect this data. This could be done in a future study.

Results:

Comment on study sample: You calculated 86 participants were needed, however assessed only 81 for eligibility. Why did you stop at 81?

Great point. We screened only 81 participants because only 81 people reached out to the primary investigator with an interest to participate in the study. We have clarified this in the text as well (lines 206-207).

Comment on participants characteristics: what other categorical variables could influence the study results? Did you collect data on for example status (single, partner etc), living arrangement (with parents, alone, etc), socioeconomic indicators, the use of other (asthma) medications?

Great thoughts. We did not collect any additional data outside of gender, ethnicity, race, smoking status, length of asthma diagnosis, and duration of days participant missed work or school. 

Consider presenting results of ANOVA in a table in addition to the text presentation.

The ANOVA results have been presented in tables. Another reviewer suggested deleting the text presentation which we followed. Thank you.

Figure 2: consider explaining MID in the capture, making it convenient for the reader to understand the figure without referring to the manuscript text.

We have made the suggested changes, thank you (lines 310-311).

Discussion

Line 304-311: very interesting discussion, and might indicate a need for eHealth intervention that allows for engaging the participants, e.g., interactive responses? Are there other asthma mediation eHealth interventions targeting the young adults? Maybe you could extend your discussion including results from such studies? For example van Buul, A.R. et al.  2020. eHealth only interventions and blended interventions to support self-management in adolescents with asthma: A systematic review. Clinical eHealth;  49-62, ISSN 2588-9141, https://doi.org/10.1016/j.ceh.2020.06.001.

Great point. The suggested study was cited and discussed as recommended. [line 359 – 361]

Line 334-343 and 371-375: Regarding adherence to asthma medications, I was wondering if the study participants had a challenge with adherence from baseline? Were they nonadherent? I guess it would be even more difficult detecting significant changes in adherence if the participants are adherent from baseline. Maybe future research should target a population with adherence challenges?

This is a good point. I have included the statement below in the discussion section to buttress your point (lines 365-367).

Furthermore, our participants were adherent to their ICS at baseline making it difficult to detect the impact of the intervention. Future studies should target a nonadherent patient population.

Comment on asthma control (from line 353): it would be interesting to have data on the use of SABA as an indicator of asthma control. Maybe add some discussion on additional clinical outcome measures for future research?

Great suggestion, we have added a discussion on using SABA as an indicator of asthma control (lines 426-428).

Limitations

With reference to previous comments on randomization and blinding: please add information on possible selection bias, performance bias, and detection bias. I recommend using RoB 2 as a guide for evaluation (https://methods.cochrane.org/bias/resources/rob-2-revised-cochrane-risk-bias-tool-randomized-trials).

Thank you for this feedback. We do not believe there was a risk of selection bias because we utilized a randomized controlled trial design for the study. Participants were randomly assigned to receive either gain or loss framed messages using an online computer program. Due to the nature of intervention (remote) and method of data collection (self-report via survey link sent to phone), performance and detection biases are not applicable.

References:

The reference list needs thorough revision; many references are missing.

Thank you for the feedback. Our reference list has been revised and updated.

Round 2

Reviewer 2 Report

Comments and Suggestions for Authors

This paper is an important contribution and I recommend that it be accepted for publication.

Reviewer 3 Report

Comments and Suggestions for Authors

New version is improved.